# Feasibility and Adoption of a Focused Digital Wellness Program in Older Adults

**DOI:** 10.3390/geriatrics6020054

**Published:** 2021-05-19

**Authors:** Eric Tam, Pedro Kondak Villas Boas, Fernando Ruaro, Juliane Flesch, Jennifer Wu, Amelia Thomas, James Li, Felipe Lopes

**Affiliations:** 1Department of Family and Community Medicine, University of California-San Francisco, San Francisco, CA 94110, USA; 2Mighty Health Digital Technologies, 5214F Diamond Heights Blvd #554, San Francisco, CA 94131, USA; pedro@mightyhealth.com (P.K.V.B.); fernando@mightyhealth.com (F.R.); juh@mightyhealth.com (J.F.); jenn@mightyhealth.com (J.W.); amelia@mightyhealth.com (A.T.); james@mightyhealth.com (J.L.); felipe@mightyhealth.com (F.L.)

**Keywords:** digital health, wellness program, geriatric mental health

## Abstract

Digital health programs offer numerous psychological and physical health benefits. To date, digital programs have been aimed broadly at younger participants, yet older individuals may also benefit. Our study sought to demonstrate user feasibility and satisfaction in a digital wellness program for older adults. We conducted a retrospective analysis of 140 participants in a digital health wellness application that integrated guided exercises, nutrition planning and health education. Primary outcomes were active participant retention, engagement in the mobile program and user satisfaction as operationalized by NPS scores. Among 140 participants, median age was 59.82 (50–80), 61% female, in a sample taken in the United States. Engagement was high and sustained, with more than 65% participants engaged, operationalized as at least completing one task activity a month over 17 weeks. Participants were also satisfied with the program, reporting NPS scores of 43 on day 30 of the program. Secondary health outcomes included 3.44 pound weight change during the first month. User feasibility and satisfaction was demonstrated in a sample of older participants for this novel digital health wellness program. Future work focused on older adult users may result in improvements in patient health outcomes and improved preventive medicine strategies.

## 1. Introduction

Over the last decade, digital programs aimed at patients with chronic diseases and overall physical and psychological well-being have becoming increasingly common and popular [1]. Within the past 10 years, the percentage of US adults owning a smartphone has increased from 35% to >80% [2], creating a digital environment conducive to rapid transmission of information and continuous communication. This digital environment has created a unique opportunity for rapid data-driven testing and dissemination of effective solutions to support preventive medical strategies and to encourage the adoption of healthy psychological and physical lifestyles. Digital therapies (e.g., telehealth platforms) and mobile mental health apps offer the ability to respond and reach individuals over great distances, with minimal mobility requirements. The health benefits of such digital programs for improving the outcomes of patients with chronic medical conditions such as diabetes, cardiovascular disease and weight loss have been well established in a diverse range of participants across the age spectrum [3,4]. Regular physical activity amongst older adults has been associated with improvements not only in general health such as blood pressure reduction [5] and weight loss [6], but also a reduction in overall cardiovascular and mortality risk [7]. These effects may also result in improvements in social cognition and attention [8]. This encouraging early data on physical and psychological well-being amongst older adults carries potential implications for wellness programs aimed at lifestyle modification and preventive medicine within the general population. Paralleling the rise in digital technologies, there has been a rise in the number of such wellness based programs [9] which has been accelerated by the events of the ongoing COVID-19 global pandemic [10]. Patients and providers encounter a dizzying number of possible digital solutions and services. For example, there are currently over 10,000 mental health related smartphone apps alone, each with diverse approaches ranging from remote cognitive behavioral therapy, to the use of automated “chatbots” driven by data science [11]. Despite this promising growth of digital health, less work has been focused on the important and growing aging population (e.g., adults over the age of age 55). Most of these programs are often designed largely by younger developers and clinicians. As a result, they have been aimed broadly at a younger working age demographic [12], given the assumption that this demographic is more engaged with technology when compared to older adults. Older individuals represent an important and large proportion of the general population for whom digital health strategies may be particularly beneficial. Past work has already noted the positive impact of digitally based strategies for older patients in diverse domains from mental health to physical balance and recovery disease [13,14]. However, numerous potential challenges exist for older adults, including physical limitations (e.g., vision, memory, cognitive impairments) [15], concerns about decreased overall familiarity/facility with technology [16], user hesitation with adoption of novel technology [17], and finally programming/content aimed for a young target demographic. Previous work has already noted that older patients are both less likely to use digital health tools, and more hesitant in adopting such measures [18,19]. Left unaddressed, this “digital divide” across the age spectrum has important consequences for exacerbating existing health care disparities amongst older patients, as well as reducing opportunities for older patients to engage in beneficial mobile interventions. Digital programs focused on older adult from an early design stage must incorporate the unique needs and challenges of this large demographic. This may allow for increased user retention and satisfaction. Therefore, our study aimed to describe user feasibility and satisfaction of a digital wellness program in a sample of older adults.

## 2. Materials and Methods

### 2.1. Study Design and App Design

This was a retrospective study of anonymized and deidentified data on participants who were enrolled in Mighty Health, a mobile based health wellness program with an older-aged focused program integrating physical exercises, coach support, and health lifestyle strategies. The program encompasses a mobile app with participants encouraged to create and follow a personalized well-being plan including a virtual coach and structured lesson and weekly goals aimed at guided exercises for improving physical health, nutrition, and general health habits such as sleep hygiene and mental health. Age specific efforts to increase usability and interest/engagement amongst older participants included larger font size, high contrast font color text and large tap areas, joint-friendly exercises, cardiovascular/diabetes focused nutrition plans, and general health content material for the older population. Inclusion criteria included users of Mighty health from the dates 10 June 2019 to 25 October 2020 over the age 50, who completed at least two tasks weekly (out of a possible average of four tasks per week) in the first month of the program. For measuring inclusion and user engagement we followed past investigations in the digital health literature [20,21,22], namely, the number of times users opened the mobile app per week/month and completed one task lesson or track within the Mighty program during a 17 week duration. The protocol was reviewed and deemed to be IRB exempt by an independent third party Institutional Review Board (Pearl IRB). 

### 2.2. Study Measures

Descriptive demographic data was collected. Primary outcomes were user retention and overall satisfaction with the program within a 4 months (17 weeks) timeframe. Participant satisfaction was assessed using net promoter scores (NPS), a management tool popular in industry that assesses customer experience and satisfaction and examines the percentage of participants rating their likelihood of recommending the service/product to a friend or colleague [23,24].

## 3. Results

There were 150 subscribers in the digital program, of which 140 ultimately met inclusion criteria for this analysis (Figure 1).

Participants characteristics are summarized in Table 1. The median age for participants included in the analysis was 60.00 (with a mean of 59.82), with 61% self-identified as female. Participants on average completed 172 tasks. At 4 months (17 weeks) 65% of users were still using the program. Finally, satisfaction as assessed by NPS scores at 30 days was 40 overall by providers, and 43 amongst individuals over the age of 50. 

No sex differences were seen between male and female participants with regards to retention (X^2^ = 0.29, *p* = 0.59) and overall satisfaction (X^2^ = 1.81, *p* = 0.18). Additionally, when stratified by age groups (50–64, and 65–80), no significant differences were seen with participants stratified by age (X^2^ = 0.84 *p* = 0.359) and overall satisfaction (X^2^ = 2.58, *p* = 0.28).

While not a primary outcome of our feasibility study, on average participants in the program completed 412,295.46 steps. Additionally, of the 140 participants 19 had weight inputs in the first and last month. For these, the average weight loss was 3.44 pounds.

## 4. Discussion

Our study found that older patients were successfully engaged in a general wellness mobile app and reported high degrees of satisfaction with the usability and overall program. This finding was seen both in men and women, and even in participants in the upper range of our sample with regards to chronological age.

Older individuals represent an important and increasingly large proportion of the general population. Digital health and wellness programs offer the potential for an amplification and increased facilitation of support and healthy preventive strategies in the general population, but older patients encounter numerous challenges with regards to potential usability and retention. Our findings on older adult adherence with regards to mobile adoption were similar to adherence rates previously published on the general population, arguing for the notion that older adults represent a particularly promising group of patients who may benefit from the design and implementation of digital programs aimed at their demographic [3]. The use of a digital health platform that included features that integrated both health recommendations and accountability (e.g., via a behavioral coach) may be a unique feature encouraging patient motivation and accountability. Additionally, having a user experience specifically aimed at older adults, from the large tap areas and font size, to the joint friendly exercises and specific health content material, may have helped drive participants to experience such sustained engagement with the specific Mighty health app used in this study. Future work should build on the notion of a “user-centric” approach, leveraging both qualitative data (e.g., focus groups and survey work) in addition to other data driven approaches to identify key aspects of mobile programs that present challenges, in addition to attractive features that would be of most interest/utility to older users. Additionally, a reconceptualization of the network of older adults may offer important future opportunities for synergy and additional support. For many older adults, both primary and extended family play an important role in overall physical and psychological support [25]. Building programs that integrate both the primary user as well as their broader social network may lead to additional areas of engagement and enjoyment. 

Many of the digital offerings today both in the private and public marketplace should also emphasize broader integration, complementing existing medical system infrastructure. The data provided by such digital applications may provide primary care providers and gerontologists with additional critical data with regards to activity and behavior that may lead to adjustments in treatment strategies or approaches [26]. Future work may also explore the implementation of such digital health strategies in a range of older adults, to identify what unique challenges may be present for users, including a diverse range of social determinants of health such as socioeconomic, primary language, and race/ethnicity, Additionally, programs should focus on adapting content to specific target demographics and avoiding a “one size fits all approach” to digital health solutions. There has been a rapid rise in the use of remote telehealth platforms within the healthcare industry over the last few years. Understanding the relationship of mobile programs to the existing innovative approaches seen in healthcare may lead to unique synergies and new innovative approaches to preventive care. Finally, prospective studies may explore the association between digital wellness programs and their impact on near and long term health outcomes such as cardiovascular risk and medication adherence. 

Our study had several limitations. First, while this study was broadly inclusive of all interested participants in the mobile platform, recruitment was done via active users for the program, so a selection/motivation bias may have been present. Additionally, as this was a naturalistic observation report, we did not have an a priori comparison group, although our comparisons to extant literature on user retention and satisfaction with mobile apps supported our conclusion that older adult users adopted the mobile technologies at similar rates [22]. Finally, as this was a pilot feasibility study, we did not explore more detailed analyses of what individual factors of the program may have contributed to retention/satisfaction amongst participants. Future work examining this, in addition to health outcomes as primary end points, will provide additional clarification on not only the usability and adoption of such digital programs but their impact on physiological and psychological well-being amongst users.

## 5. Conclusions

In conclusion our study found that older adults had high and sustained engagement in a digital wellness app/program. Our work suggests that digital health and wellness programs should not be limited to younger age demographics and can be adopted by older adults. The expansion of a breadth of digital healthcare products for the general population offers the promise of improving the adoption of healthy lifestyles and behavior that can ultimately positively impact user well-being. Older individuals in particular may stand to benefit from such programs. Moving forward, it is vital to put forth a program of work aimed at understanding the unique challenges and needs of older individuals, particularly within the digital health sphere. By doing so, clinicians, researchers, and industry may help bridge any potential digital divide and provide broad based, tailored interventions to improve care and patient well-being across the age spectrum.

## Figures and Tables

**Figure 1 geriatrics-06-00054-f001:**
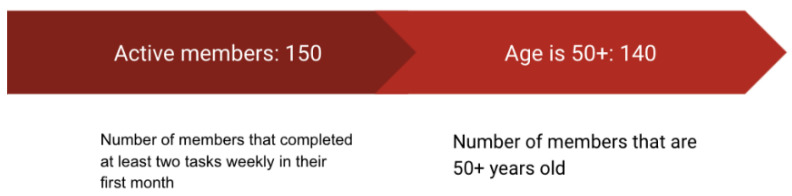
Participant Flow Chart.

**Table 1 geriatrics-06-00054-t001:** Participant Characteristics.

Characteristic	n	Overall
Total Sample	140	
Age (in years)		Mean (59.82) median (60)
Age (standard deviation)		6.42
Age (range)		50–80
Age Mode		60
Female		85 (61%)
Male		55 (39%)
Active Users at 16 weeks (4 months)		91 (65%)

## Data Availability

Data supporting reported results can be found on dataset stored by Mighty Health Incorporation.

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
