# Peer review of "Feasibility and Adoption of a Focused Digital Wellness Program in Older Adults"

_geriatrics, 2021, doi:10.3390/geriatrics6020054_

Round 1

Reviewer 1 Report

Original appraoch of the problem. 

Author Response

We thank the reviewers and editor for their time in reviewing our paper. 

Reviewer 2 Report

Several aspects related to guidelines for authors have not been taken into consideration. Affiliations are incomplete; abstract structure includes subsection headings; abstract length, and other items that deserve attention.

The work is done over a sample of 140 participants, ranged 50-80 years old but the analysis is done as a single age group, instead of segregated data for at least 2 age groups 50-65 and 65-80. Since this is a journal for geriatrics, specific profile for old people is expected to be reported, which is not the case. As it stands, it does not provide a clear profile and there are not control activities for comparison.
Report of results should be the center of attention and is reduced to a minimum.
Age can not be rounded up. Mean age 59.82 can not be rounded up to 60.0. Also, please, note that the use of 'elderly' can be considered as ageist.

Author Response

We thank the reviewers and editor for their time in reviewing our paper. We believe our manuscript is significantly improve due to the comments/insight from the reviewers and we appreciate their time. We have attached responses addressing reviewer comments, point by point. Please let us know if you need any additional information. Thank you again for your time!

Several aspects related to guidelines for authors have not been taken into consideration. Affiliations are incomplete; abstract structure includes subsection headings; abstract length, and other items that deserve attention.

Thank you for these comments. We have updated complete affiliations.

-Abstract length was 275. We now have trimmed it to 199 and removed subheadings as requested.  

-References are now in brackets per editor’s request

The work is done over a sample of 140 participants, ranged 50-80 years old but the analysis is done as a single age group, instead of segregated data for at least 2 age groups 50-65 and 65-80. Since this is a journal for geriatrics, specific profile for old people is expected to be reported, which is not the case. As it stands, it does not provide a clear profile and there are not control activities for comparison. Report of results should be the center of attention and is reduced to a minimum.

We thank the reviewer for this insightful remark. We now include an additional analysis with the sample of participants stratified by age (2 categorical groups of aged 50-64, 65-90). In our analysis we found that the two age groups did not differ significantly with regards to adherence or satisfaction. (please see comment from earlier reviewer): When stratified by age groups (50-64, and 65-80), no significant differences were seen with participants stratified by age (X2=0.84 p=.359) and satisfaction.

Age can not be rounded up. Mean age 59.82 can not be rounded up to 60.0.

Thank you for this comment. We now clarify that we included the mean age of 59.82 with a median age of 60

Also, please, note that the use of 'elderly' can be considered as ageist.

We apologize for the inclusion of the word “elderly”. We agree and understand the perspective from the reviewer with regards to potential for negative implications re: ageism. We have updated the text to reflect this.

Reviewer 3 Report

The topic of the manuscript is of interest in the field of elderly populations and geriatrics. The study investigated whether older people were or not engaged in a wellness mobile app and investigated degree of satisfaction with the app.

Despite this is really of interest, particularly in times of pandemic, several changes should be made prior to the publication. I will try to make some suggestions by following sections and subsections of the manuscript.

In the abstract section, the authores reported they carried out an observational study including 140 individuals. In the body of the manuscript, they indicated that this was a retrospective study. I would recommend to indicate it in the abstract section. Which variables should be considered retrospective and which of them prospectively recorded?

In the introduction section, the authors revise the role of technologies in older people accross the age spectrum. Lifetyle modifications and preventive actions are highlighted. I would expand this part at the introduction section, by focusing the introduction on the relationship between age, physical exercise and cognitive performance. Which is the effect of physical exercise, and by implication, of the use of technologies, in the clinical evolution of cognition?

Statistical analysis are not very complex. The methods are really good and the topic is really of interest for scientists working in the field. I would add a few more complex analysis. Was retention statistically significant between men and women? In other words, why do not the authors create a new subsection in the results section to present results according to gender. Are there any  gender differences in the satisfaction?

Which advantages may  technologies used in the paper have compared to the other ones? Why is this option better than the others? 

A section about future perspectives should be better developed.

Author Response

We thank the reviewers and editor for their time in reviewing our paper. We believe our manuscript is significantly improve due to the comments/insight from the reviewers and we appreciate their time. We have attached responses addressing reviewer comments, point by point. Please let us know if you need any additional information. Thank you again for your time!

The topic of the manuscript is of interest in the field of elderly populations and geriatrics. The study investigated whether older people were or not engaged in a wellness mobile app and investigated degree of satisfaction with the app. Despite this is really of interest, particularly in times of pandemic, several changes should be made prior to the publication. I will try to make some suggestions by following sections and subsections of the manuscript.

Thank you. We appreciate it.

In the abstract section, the authors reported they carried out an observational study including 140 individuals. In the body of the manuscript, they indicated that this was a retrospective study. I would recommend to indicate it in the abstract section. Which variables should be considered retrospective and which of them prospectively recorded?

We have adjusted the wording for uniformity for study design clarification. The data was a retrospective analysis of an existing dataset. We now state throughout the manuscript that this was a retrospective study of an existing dataset including in the abstract for clarity.

In the introduction section, the authors revise the role of technologies in older people across the age spectrum. Lifestyle modifications and preventive actions are highlighted. I would expand this part at the introduction section, by focusing the introduction on the relationship between age, physical exercise and cognitive performance. Which is the effect of physical exercise, and by implication, of the use of technologies, in the clinical evolution of cognition?

Excellent point. We have now expanded this introduction section to include a discussion of past work relating age, physical exercise, and cognitive performance. Additionally, we now cite some literature regarding physical exercise, digital health, and clinical evolution of cognition. Please see Introduction with additional citations:

  1. Kelley, G., & McClellan, P. (1994). Antihypertensive effects of aerobic exercise: a brief meta-analytic review of randomized controlled trials. American Journal of Hypertension7(2), 115-119.
  2. Valencia, W. M., Stoutenberg, M., & Florez, H. (2014). Weight loss and physical activity for disease prevention in obese older adults: an important role for lifestyle management. Current diabetes reports14(10), 539.
  3. Buchner, D. M. (2009). Physical activity and prevention of cardiovascular disease in older adults. Clinics in geriatric medicine25(4), 661-675.
  4. Brawley, L., Rejeski, W. J., Gaukstern, J. E., & Ambrosius, W. T. (2012). Social cognitive changes following weight loss and physical activity interventions in obese, older adults in poor cardiovascular health. Annals of Behavioral Medicine44(3), 353-364

Statistical analysis are not very complex. The methods are really good and the topic is really of interest for scientists working in the field. I would add a few more complex analysis. Was retention statistically significant between men and women? In other words, why do not the authors create a new subsection in the results section to present results according to gender. Are there any gender differences in the satisfaction?

We have updated our analytic plan to now explore any potential effects by sex/gender in addition to age. We have included a new subsection in the results section exploring gender differences with regards to adherence and satisfaction. Additionally, we have also included another subsection exploring differences by age (50-64 and 65 and older). In summary, no sex differences were seen between male and female participants with regards to retention (X2 = 0.29, p=.59) and satisfaction. Additionally, when stratified by age groups (50-64, and 65-80), no significant differences were seen with participants stratified by age (X2=0.84 p=.359) and satisfaction. Our sub-analysis suggests that our digital health program was associated with high rates of adherence and satisfaction even adjusted for age/sex. 

Which advantages may technologies used in the paper have compared to the other ones? Why is this option better than the others? 

This is a great question and something we have now expounded on in the updated discussion section. Briefly, we discuss how unlike other interventions which are geared towards a broad demographic, the focus of an intervention designed around not just evidence based recommendations for physical activity but building in a component of motivation and accountability (via a coach) may help improve user experience compared to other options currently existing without such feature. Additionally, the technology (e.g. digital app) described in this study was aimed primarily at older adults. Age specific efforts to increase usability and interest/engagement amongst older participants included larger font size, high contrast font color text and large tap areas, joint-friendly exercises, cardiovascular/diabetes focused nutrition plans, and general health content material for the older population were done.

A section about future perspectives should be better developed.

We now fleshed out the future perspectives section in the update manuscript with a discussion of implications for cardiovascular and metabolic health, in addition to linkage to existing primary care management.

Round 2

Reviewer 2 Report

The authors have taken care of the previous version, but still some minor aspects have been left (i.e. references 5,6,7,8)

The segregation of data in two age ranges has allowed to conclude that both age groups are not different with respect to the variables analyzed. This is an interesting result.

Please, note, that the results are still providing a glance on the data and some reports of variables results are from a specific part of the sample, or specific time points, etc.  
A table is needed with the results depicted or the text should provide a more detailed description of variables measured, part of the sample that is included in that analysis and clear results (mean + SEM or SD, or percentages).

In the abstract, the sociodemogeographic report refers to 39 states within US. This data is in disagreement with that reported in the results, when the authors refer to "a broad geographic range within the Continental United States and 7 other countries."

what does "on this im-23 portant demographic" means?

Please, indicate how the app verified that the user was the same participant.

The authors refe to ' general health habits such as sleep hygiene and mental health' but no mention on how the benefits were measured. Which scales, etc Please, report in the methodology part and provide the corresponding results.

In order to report 'active participant retention' and how the satisfaction refers to the whole samples, a flow chart of the initial sample and final participants included in the analysis is needed. The program involves out of a possible average of 4 tasks weekly, and those who at least completed an averaged of 2 in the first month were included.

In the abstracts, however, there's a discrepancy with the above mentioned results, as it is indicated "operationalized as at least completing one task activity a month over 17 weeks."

 In the part of the discussion to refer to the limitations, which is an important exercise of scientific auto-analysis the authors use an incorrect form to report as the sentence refers to how strength was limited. "The strength of our conclusions was limited by numerous variables."  Please, note, scientific community expects to address limitations with a clear statement that reflects the weak points, the reasons they could not be avoided and the way the authors could remend in the next experimental approaches.

Conclusions provide a summary view of the relevance of digital healthcare and a perspective of this promising area but this paragraph should better belong to the introduction. Please, provide a conclusion summarizing the conclusions of the present data report and their implications.

Author Response

Please see the attachment and associated figures and tables
